# FBB18 participates in preassembly of almost all axonemal dyneins independent of R2TP complex

Limei Wang[1] , Xuecheng Li[1] , Guang Liu[1], Junmin Pan [1,2] *

1 MOE Key Laboratory of Protein Sciences, School of Life Sciences, Tsinghua University, Beijing, China,
2 Laboratory for Marine Biology and Biotechnology, Qingdao National Laboratory for Marine Science and Technology, Qingdao, Shandong Province, China

⊜ These authors contributed equally to this work.
* panjunmin@tsinghua.edu.cn

**Data Availability Statement:** All relevant data are within the manuscript and its Supporting Information files.

**Funding:** This work was supported by the National Natural Science Foundation of China (31991191,

## Abstract

Assembly of dynein arms requires cytoplasmic processes which are mediated by dynein preassembly factors (DNAAFs). CFAP298, which is conserved in organisms with motile cilia, is required for assembly of dynein arms but with obscure mechanisms. Here, we show that FBB18, a *Chlamydomonas* homologue of CFAP298, localizes to the cytoplasm and functions in folding/stabilization of almost all axonemal dyneins at the early steps of dynein preassembly. Mutation of FBB18 causes no or short cilia accompanied with partial loss of both outer and inner dynein arms. Comparative proteomics using [15]N labeling suggests partial degradation of almost all axonemal dynein heavy chains (DHCs). A mutant mimicking a patient variant induces particular loss of DHCα. FBB18 associates with 9 DNAAFs and 14 out of 15 dynein HCs but not with IC1/IC2. FBB18 interacts with RuvBL1/2, components of the HSP90 co-chaperone R2TP complex but not the holo-R2TP complex. Further analysis suggests simultaneous formation of multiple DNAAF complexes involves dynein folding/stability and thus provides new insights into axonemal dynein preassembly.

## Author summary

Motile cilia are important for human physiology and defects in cilia motility may cause human disorders such as male infertility and primary ciliary dyskinesia. The motility of cilia requires preassembly of axonemal dyneins. Using a combination of genetic and other approaches, we have studied the working mechanism of *FBB18*, a *Chlamydomonas* homologue of *CFAP298*, defects in which result in primary ciliary dyskinesia. We found that FBB18 participates in dynein folding/stability in the cytoplasm, which is distinct from its proposed function in ciliary targeting of dynein complexes or stabilization of dynein arms within cilia. In addition, we have provided evidence that multiple distinct complexes are simultaneously formed to participate in dynein folding, thus providing new insights into dynein preassembly. Last but not least, we showed that RuvBL1/2 of the HSP90 co-chaperone R2TP complex may function independently of the R2TP complex in dynein disassembly. This work has both scientific and medical significance and will be of general interest to the fields of ciliary biology and protein folding/stability.

31972888) (https://www.nsfc.gov.cn/), by the Pilot National Laboratory for Marine Science and Technology (MS2019NO04) (http://www.qnlm.ac) and by the National Key R&D Program of China (2017YFA0503500) (https://www.most.gov.cn) to JP. The funders had no role in study design, data collection and analysis, decision to publish, or preparation of the manuscript.

**Competing interests:** The authors have declared that no competing interests exist.

## Introduction

Motile cilia and flagella (exchangeable terms) are highly conserved throughout eukaryotes. The beating of a cilium provides forces for cell swimming in unicellular organisms. Its role is expanded in mammals. In addition to propelling sperm movement, it functions in circulation of cerebrospinal fluid in the brain, mucus clearance in airways, egg movement in fallopian tubes, sperm transport from testis to epididymis and nodal flow in embryonic node. Ciliary beating requires axonemal dyneins, which consist of heavy chains (HCs), intermediate chains (ICs), light intermediate chains (LICs) and light chains (LCs) while the exact composition of each dynein differs [1, 2]. The axonemal dyneins are organized into outer and inner dynein arms (ODAs and IDAs) as revealed by electron microscopy. ODAs from *Chlamydomonas*, a unicellular organism widely used for ciliary studies, are three-headed dyneins that are composed of three HCs (DHCα, β, γ), a pair of IC1/IC2 and several other subunits. IDAs include one two-headed dynein and 10 monomeric dyneins. A total of 15 axonemal dynein HCs including three outer arm dynein HCs and 12 inner arm dynein HCs are present in *Chlamydomonas* while mammals have 5 outer and 8 inner arm dynein HCs [3, 4]. Mutations in dynein subunits impair ciliary motility, leading to primary ciliary dyskinesia characterized by chronic respiratory infections, male infertility and laterality defects [5].

The dynein arm assembly requires preassembly of the dynein complexes in the cytoplasm before they are transported to cilia and docked on the ciliary axoneme (see review [6]). In the cytoplasm, dynein HCs as well as ODA ICs are properly folded and stabilized, assembled into a complex with other dynein subunits followed by maturation, packaging and targeting to cilia. More than a dozen factors have been shown to be dedicated to these processes and termed dynein, axonemal, assembly factors (DNAAFs) [2,6–8]. DNAAF12/LRRC56 and DNAAF11/LRRC6 are predicted to mediate later steps of dynein preassembly or transport because the dynein HCs and ODA IC1/IC2 showed normal abundance in the cytoplasm with loss of dyneins in cilia [9–11]. DNAAF9/Shulin packages outer arm dynein complexes [12] and DNAAF18/ODA16 serves as a cargo adapter of outer arm dynein complexes for ciliary targeting via the intraflagellar transport machinery [13]. The other factors appear to be critical for folding and stabilization of dynein HCs which is reflected by reduced abundance upon mutations in DNAAFs [6, 8, 14] while ZMYND10 is also involved in stability of IC1/IC2 of ODA [14]. How these factors interact and cooperate to mediate folding and stabilization of dyneins remain obscure.

Chaperone HSP90 functions in the folding, stability and assembly of a wide array of protein complexes [15]. Several DNAAFs have been found to associate with HSP90 by immunoprecipitation [8], suggesting that DNAAFs function in recruiting dynein clients to HSP90. DNAAFs that directly interact with HSP90 from experimental evidence or prediction include TPR domain containing proteins DNAAF4/DYX1C1, DNAAF13/SPAG1 and its paralog RPAP3 [16]. DYX1C1 interacts with DNAAF2/PF13 or DNAAF6/TWI1 [16, 17]. SPAG1/RPAP3 are homologues of yeast Tah1, a component of a R2TP complex (Rvb1, Rvb2, Tah1 and Pih1 in yeast and RuvBL1, RuvBL2, RPAP3 and PIH1D1 in mammal), which is a HSP90 co-chaperone [16, 18, 19]. Mutations in the components of R2TP or its like complex in different organisms lead to instability of axonemal dyneins [16, 20–27]. In addition, DNAAF10/WDR92 interacts with the R2TP complex in vivo and it thus has been proposed that DNAAFs also recruit the dynein clients to HSP90 via the R2TP complex for folding and assembly [23, 24]. RuvBL1/2 may function independent of R2TP in the assembly of other protein complexes and have been proposed to possess chaperone activities [28]. However, it is not clear whether RuvBL1/2 alone may function in dynein folding and assembly.

CFAP298 (aka C21orf59) is a conserved protein among organisms with motile cilia [29, 30]. Mutations in human patient and zebrafish and RNAi depletion in planaria cause loss of both ODAs and IDAs [31, 32]. It has been proposed that CFAP298 functions in cilia to enhance ODA stability in addition to possible roles in cytoplasmic preassembly and transport of dynein complexes. However, no experimental data has been shown for its working mechanism in the cytoplasm. In this work, by studying FBB18, a *Chlamydomonas* homologue of CFAP298, we found that FBB18 is involved in folding and stabilization of almost all outer and inner dynein HCs but not in ciliary targeting or stabilization ODAs within cilia. FBB18 forms an interacting network(s) with other nine DNAAFs. More importantly, this study shows that multiple DNAAF complexes are simultaneously formed during folding and stabilization of dynein HCs and RuvBL1/2 may function independently of a R2TP complex in dynein preassembly.

## Results

### An *fbb18* mutant shows defects in ciliogenesis and ciliary motility

FBB18, a *Chlamydomonas* homologue of CFAP298, is a protein of 277 residues with two conserved domains: a coiled-coil domain and a domain of unknown function DUF2870, which are also present in its human homologue (Fig 1A). We identified an *fbb18* mutant (*fbb18-1*) in

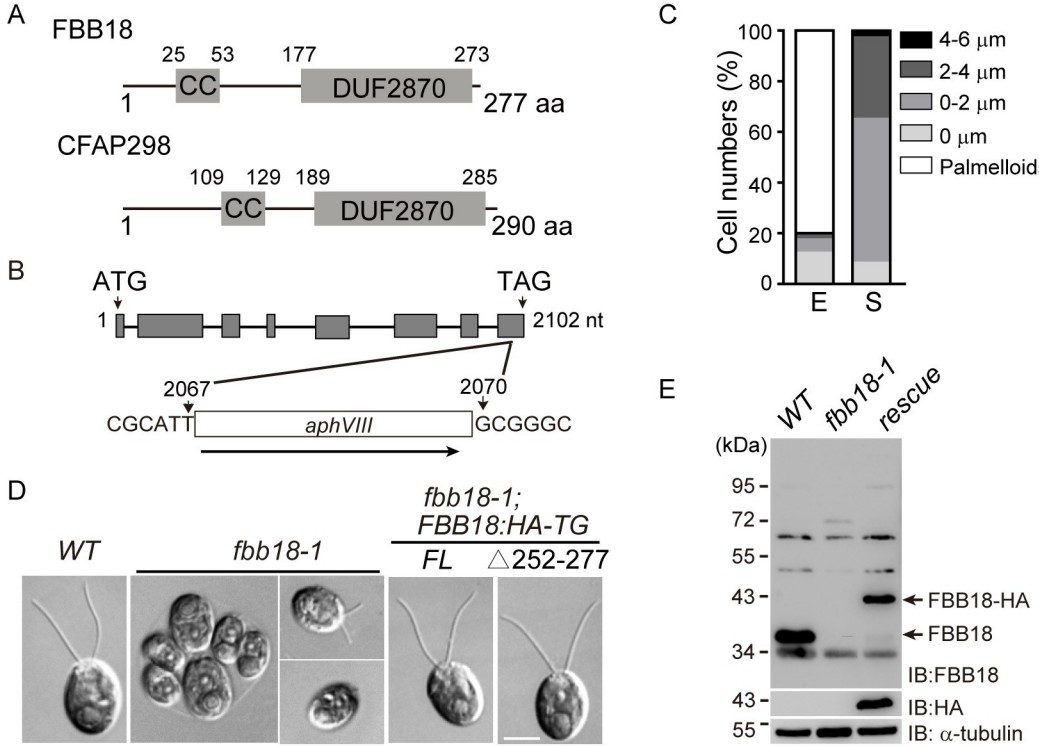

**Fig 1. Isolation and characterization of an *fbb18* mutant.** (A) Diagrams showing the domain structures of FBB18 and its human counterpart CFAP298. The numbers indicate the positions of the residues. CC, coiled-coil domain. (B) Schematic illustration of the *FBB18* gene shows the insertion of a foreign DNA fragment containing paromomycin resistance gene *aphVIII*. The insertion was identified by PCR and DNA sequencing. Grey box, exon; line, intron; numbers, nucleotide positions. (C) Ciliary phenotypes of *fbb18-1* grown in exponential (E) and stationary (S) growth phases. (D) DIC cell images show the ciliary phenotype of wild-type (*WT*), *fbb18-1*, transformants expressing full-length (*FL*) or truncated (Δ252–277) gene. Bar, 5 μm. (E) Immunoblot analysis of wild-type (*WT*), *fbb18-1* and rescued cells expressing *FBB18*: *HA-TG* with the indicated primary antibodies.

which the 2068–2069 nt in the last exon is replaced by a foreign DNA insert (Fig 1B). In contrast to wild-type strain, which is unicellular with two cilia of about 12 μm in length, most of the cells in *fbb18-1* formed multicellular aggregates known as palmelloids or sporangia when grown in exponential growth phase. Cells were hatched from sporangia from unknown reason when grown at stationary growth phase and had no or short cilia (Fig 1C and 1D). The cells with cilia were immotile as revealed under microscopic observation (see also Fig 2). Thus, the mutant is defective in both ciliogenesis and motility.

FBB18 antibody recognized a band around the predicted molecular weight of FBB18 in the wild-type cells but not in the mutant, indicating that *FBB18* expression was disrupted (Fig 1E). Expression of a full-length *FBB18* tagged with HA (*FBB18:HA-TG*) in *fbb18-1* rescued the ciliary defects of the mutant in motility and ciliary length (Fig 1D and 1E). 23% of the transformants from at least three independent transformations showed normal ciliary phenotype, indicating that the rescue was caused by incorporation of the transgene and not by random disruption of some other genes. Thus, we have identified a null mutant of *FBB18* with defects in both ciliogenesis and ciliary motility.

## Null mutation of *FBB18* causes partial loss of ODAs and IDAs

*CFAP298* mutation in human or RNAi depletion of its homologues in zebrafish and planaria respectively alters cilia motility with loss of ODAs [31, 32]. Consistent with this, transmission electron microscopy (TEM) showed that *fbb18-1* cilia exhibit partial loss of both ODAs and IDAs (Fig 2A and 2B).

To further confirm the dynein arm defects in the cilia of *fbb18-1* mutant, we examined two ODA heavy chains (ODAα and ODAβ) and three IDA heavy chains (dynein b, c and e) by immunoblotting. Isolated cilia from wild-type, rescue, *oda1*, *oda5* and *pf22* cells were used as controls. *ODA1* encodes a component of ODA-docking complex (ODA-DC) and mutation of which induces loss of ODA [33]. ODA5 is involved in cytoplasmic maturation of ODAs for their binding to doublet microtubules [34]. PF22/DNAAF3 regulates the assembly of both ODAs and IDAs [7]. As expected, DHCα and DHCβ were not detected in *oda1* and *oda5* cilia while IDA components were not affected; and reductions in both ODA and IDA components were observed in *pf22* cilia (Fig 2C). In cilia of *fbb18-1*, DHCα and dynein b were undetectable while the levels of DHCβ, dynein c and dynein e were reduced compared to the wild-type and rescued cells (Fig 2C). The reduced protein levels of the dynein heavy chains are consistent with the partial loss dynein arms (Fig 2A and 2B). Thus, FBB18 is required for proper assembly of both ODA and IDA in cilia.

## FBB18$^{\Delta252-277}$ that mimics truncated mutation in human patients leads to particularly loss of DHCα in cilia

FBB18 and its human homologue CFAP298 both have a coiled-coil and a DUF2870 domain (Fig 1A). To test the function of these domains, HA-tagged FBB18 with deletion of either domain was expressed respectively in *fbb18-1* (S1A Fig). The ciliary defects of *fbb18-1* in the transformants were not rescued. We found that the expression levels of the deletion mutants were much lower than that of the wild-type gene (S1B Fig), thus preventing us to establish the functions of the deleted domains. The lower expression suggests that the mutated proteins may not be stable or the corresponding mRNAs may be subjected to decay.

A PCD patient carrying truncation mutations (p.Arg98*, p.Tyr264*) exhibits less severe defects in dynein arms than patients with homozygous truncating mutations (p.Tyr245*) [31]. Examination of the pathogenicity of the mutation (p.Tyr264*) in zebrafish was not successful because the mutated protein was expressed at extremely low level [31]. p.Tyr264* mutation

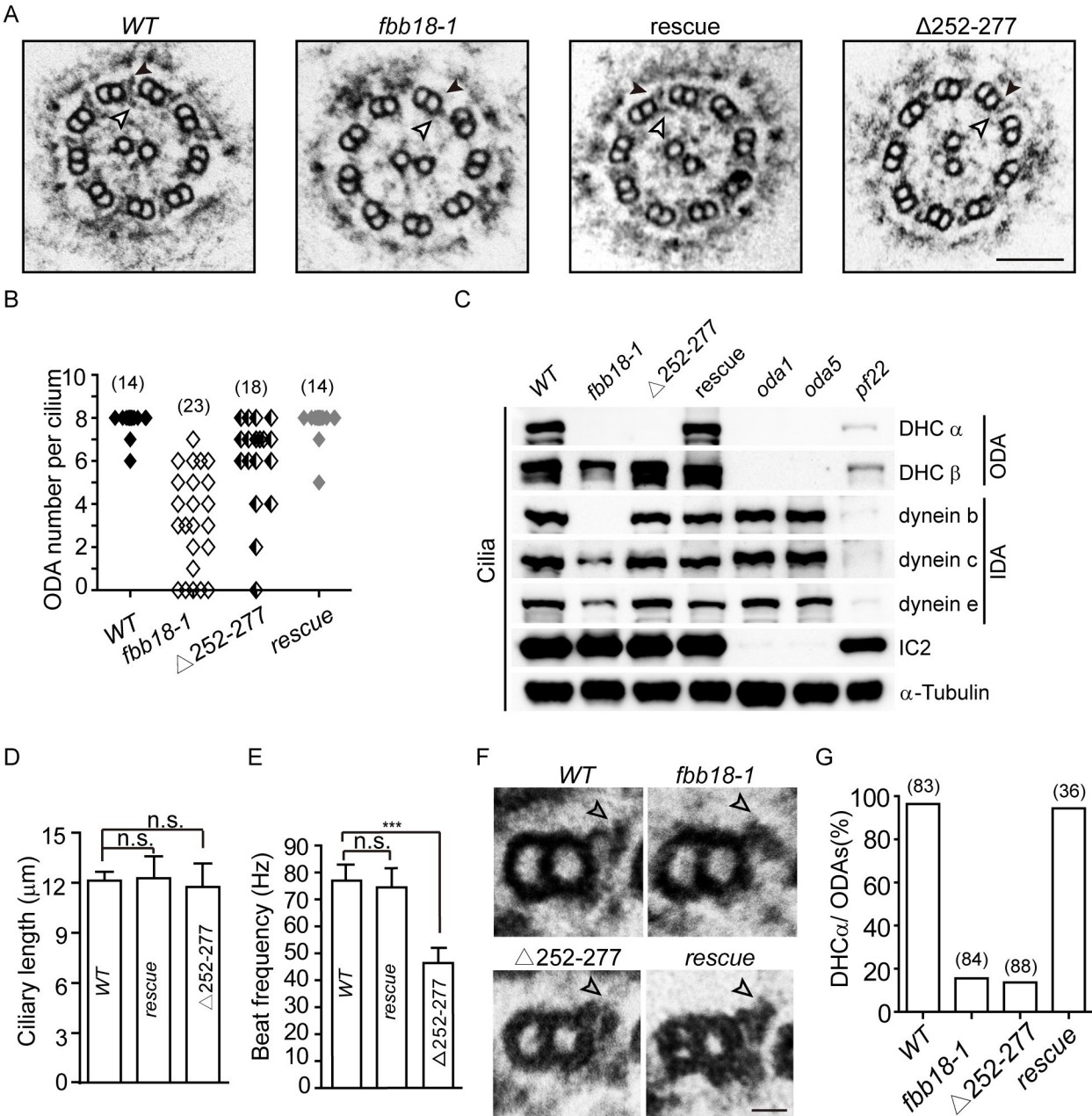

**Fig 2. Loss of dynein arms in *fbb18* mutants.** (A) TEMs of cilia cross-sections show a partial loss of axonemal outer (ODA, black arrowhead) and inner (IDA, white arrowhead) dynein arms in mutants. Bar, 100 nm. (B) Statistical analysis of ODA loss in the mutants. Numbers, TEM cross-sections examined. (C) Reduced abundance of axonemal dyneins in cilia of *fbb18* mutants. Representative axonemal dynein HCs were examined from isolated cilia of cells as indicated by immunoblotting. α-tubulin was used as a control. The abundances of dynein HCs were quantitated by densitometry. The ratios of dynein HC abundance (*wt/fbb18-1/Δ252–277*) with normalization to α-tubulin are the following: DHCα, 1.00:0.02:0.02; DHCβ, 1.00:0.73:1.00; dynein b, 1.00:0.00:0.75; dynein c, 1:0.38:0.90; dynein e, 1:0.44:1.01. (D-E) *FBB18*^Δ252–277 shows no defect in ciliary length (D) but with reduced beating frequency (D). Data were presented as mean ± SD. ns, not significant; *, p<0.05; ***, p<0.0001. (F-G) *FBB18*^Δ252–277 exhibits loss of DHCα as shown in TEM cross-section (F) and statistical analysis (G). White arrowhead shows loss of DHCα in the mutants. Bar in F, 10 nm; Numbers in G, ODAs examined.

may lead to nonsense mediated mRNA decay, it might also result in a truncated protein with disruption of the DUF2870 domain (Fig 1A). To test the pathogenicity of a truncated protein in *Chlamydomonas*, a corresponding mutant *FBB18*^Δ252−277 tagged with HA was expressed in

*fbb18-1*. The expression level of $FBB18^{\Delta252-277}$ was comparable to that of wild-type *FBB18* (S1B Fig). $FBB18^{\Delta252-277}$ mutant had normal length of cilia while the ciliary beating frequency was reduced ~40% compared to the wild-type control (Figs 1D and 2D and 2E and S1–S3 Videos). TEM examination showed that it had partial loss of dynein arms but not as severe as the *fbb18-1* null mutant (Fig 2A and 2B). Immunoblotting of isolated cilia showed that it specifically lost DHCα compared to the wild-type and the null mutant (Fig 2C). *Chlamydomonas* ODA forms a three heads architecture with DHCα on the outermost position [35]. Indeed, closer examination of the ODAs revealed that DHCα was deficient from the ODAs of $fbb18^{\Delta252-277}$ cilia as well as in the *fbb18-1* cilia (Fig 2F and 2G). These data suggest that the assembly of DHCα particularly depends on the function of the C-terminal DUF2870 domain. Whether $FBB18^{\Delta252-277}$ is a good model for studying the pathogenicity of the human variant (p.Tyr264*) depends on future verification if this mutation leads to synthesis of a truncated protein.

## FBB18 is involved in cytoplasmic preassembly of axonemal dyneins

The mRNA level of *FBB18* is immediately up-regulated after deciliation and returned to pre-deciliation level at 60 min post-deciliation [36]. Immunoblot analysis showed that the protein level of FBB18 increased at 30 min after deciliation when about half length of cilia are assembled (Fig 3A), indicating cells utilize a preexisting pool of FBB18 during earlier ciliogenesis [37]. It was intriguing how FBB18 regulates the assembly of dynein arms. It was reported that FBB18 was present in cilia with predominant localization in the ciliary membrane/matrix. In addition, its protein level increased in the cell as well as in cilia of several motility mutants of *Chlamydomonas* [31]. It was thus proposed that FBB18 functions in the stability of dynein arms in cilia and/or ciliary targeting of dynein arms. We confirmed the distribution of FBB18 in cilia (Fig 3B). However, we failed to observe significantly increased level of FBB18 in the whole cell samples of several motility mutants. The changes in the level of FBB18 in cilia from these mutants were not consistently associated with the impaired motility phenotype, which are different from a previous report [31] (S2 Fig). One possible explanation for this discrepancy is that the loading control was not shown in that study.

Immunoblotting of FBB18 in the cell body versus cilia showed that it was predominantly localized in the cell body (Fig 3C), which is consistent with the prominent presence of CFAP298 in the cell body of rat trachea ciliated cells shown by immunostaining [31]. FBB18-HA was distributed in the cell body with punctate patterns (Fig 3D), which is in contrast to the reported enrichment at the basal bodies of its mammalian homolog [31]. Whether the FBB18-HA puncta represent the dynein axonemal particles (DynAPs) as being reported in *Xenupus* remains to be tested [38].

The predominant localization of FBB18 in the cell body and loss of dynein arms in the *fbb18* mutants suggest that FBB18 is likely involved in dynein preassembly. We examined dynein heavy chains in *fbb18* mutants by immunoblotting. Unlike *oda1* and *oda5* mutants, but similar to the *pf22* mutant, both *fbb18-1* and $FBB18^{\Delta252-277}$ mutants showed decreased level of DHCα and dynein b compared to the wild-type control (Fig 3E). Thus, FBB18 likely functions to stabilize dynein HCs.

To demonstrate whether FBB18 is associated with dynein heavy chains, wild-type cell extract was subjected to immunoprecipitation with anti-FBB18 or rabbit IgG (control) followed by immunoblotting. FBB18 antibody but not IgG pulled down two axonemal dyneins that were examined (Fig 3F). Taken together, a predominantly localization of FBB18 in the cell body, its association with dynein HCs and reduced abundance of dynein HCs in the *fbb18* mutants suggest that FBB18 is a bona fide DNAAF to serve as a co-chaperone to fold/stabilize axonemal dyneins.

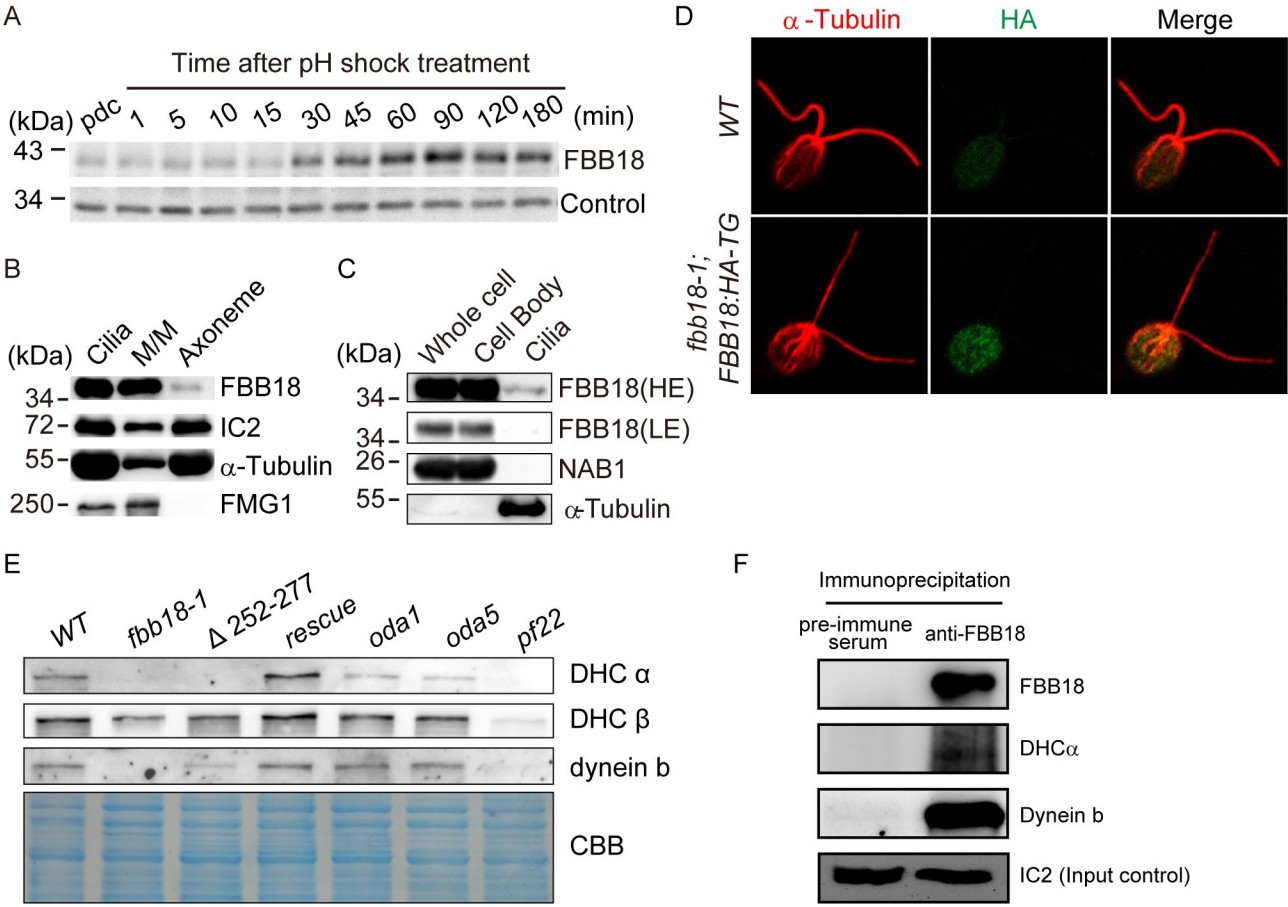

**Fig 3. FBB18 localizes predominantly to cytoplasm and mediates stability of cytoplasmic axonemal dyneins.** (A) Immunoblot analysis shows changes in FBB18 level during ciliogenesis. Cell samples from wild-type cells were taken at the indicated times after removal of cilia by pH shock. pdc, pre-deciliation. (B) Ciliary FBB18 localizes mainly to the membrane/matric fraction. Isolated cilia were fractionated into membrane/matrix (M/M) and axonemal fractions followed by immunoblotting with the indicated antibodies. (C) FBB18 is almost exclusively present in the cell body. Equal amounts of proteins from whole cell, cell body and cilia were analyzed by immunoblotting with the indicated antibodies. NAB1, a nucleic acid-binding protein, was used as a control for cell body contamination. HE, high exposure; LE, low exposure. (D) Immunostaining of cells with anti-HA and anti-α-tubulin antibodies. FBB18-HA with punctate staining patterns was observed in the cell body but barely in cilia. Bar, 5 μm. (E) Immunoblots of whole cell extracts from cells as indicated show reduced abundance of representative axonemal dyneins in the *fbb18* mutants. CCB, immunoblot membrane stained with coomassie blue showing equal loading. The normalized ratios of dynein abundance (*wt/fbb18-1/Δ252–277*) are the following: DHCα, 1.00:0.01:0.01; DHCβ, 1.00:0.55:0.99; dynein b, 1.00:0.00:0.55. (F) FBB18 associates with axonemal dyneins. FBB18 antibody or preimmune serum was used for immunoprecipitation followed by immunoblotting with the indicated antibodies.

## Differential regulation of the stability of axonemal dyneins by FBB18

After confirmation that FBB18 is indeed a DNAAF, we next determined which axonemal dynein is regulated by FBB18 by comparative proteomics. The cell extracts from $^{15}$N labelled *fbb18-1* and unlabeled rescued cells ($^{14}$N) or vice versa were mixed in an equal amount followed by SDS-PAGE. The top gel region that contains the dynein HCs was sliced for mass spectrometry analysis. All dynein HCs except four IDA minor HCs were identified (Table 1) [3]. Loss of FBB18 induced reduction in the abundance of axonemal dyneins in different degrees. For the IDA heavy chains, it resulted in 10% to 70% reductions depending on the dynein species. For the ODA heavy chains, DHCγ had no changes, DHCβ showed about 10% reduction while DHCα was reduced ~90%, which is consistent with the immunoblotting data (Fig 3E). Please note, the extent of reduction in the abundance of dyneins shown by

**Table 1. Changes in protein abundance of axonemal dyneins in *fbb18-1* versus rescued cells.**

| | Protein/Gene | *fbb18-1* labeling | | control labeling | | Average Ratio (*fbb18*/control) |
|---|---|---|---|---|---|---|
| | | PSM Count | *fbb18*/control | PSM Count | *fbb18*/control | |
| IDA HCs | DHC 1α/DHC1 | 132 | 0.91 | 107 | 0.89 | 0.9 |
| | DHC 1β/DHC10 | 70 | 0.31 | 33 | 0.46 | 0.38 |
| | dynein a/DHC6 | 48 | 0.77 | 35 | 0.86 | 0.81 |
| | dynein b/DHC5 | 33 | 0.5 | 15 | 0.42 | 0.46 |
| | dynein c/DHC9 | 92 | 0.26 | 54 | 0.32 | 0.29 |
| | dynein d/DHC2 | 52 | 0.52 | 39 | 0.70 | 0.61 |
| | dynein e/DHC8 | 19 | 0.67 | 9 | 0.53 | 0.6 |
| | dynein g/DHC7 | 73 | 0.37 | 52 | 0.43 | 0.4 |
| ODA HCs | DHC α/DHC13 | 374 | 0.09 | 235 | 0.14 | 0.11 |
| | DHC β/DHC14 | 291 | 0.88 | 240 | 0.94 | 0.91 |
| | DHC γ/DHC15 | 130 | 0.99 | 131 | 1.06 | 1.02 |
| IFT dynein | DHC1b/DHC16 | 404 | 3.75 | 450 | 2.22 | 2.98 |

The cell extracts from $^{15}$N labeled *fbb18-1* and unlabeled control cells with equal amount of proteins were mixed, or vice versa, and analyzed by SDS-PAGE and mass spectrometry.

immunoblotting does not match very well with the proteomic data, which may reflect the limitations in protein quantifications by using immunoblotting. It is intriguing that FBB18 strongly affect the stability of DHCα relative to DHCβ and DHCγ. The three HCs of ODA show high similarity in the amino sequences in the motor domains, however, the N-terminus of DHCα possesses a unique kelch-type_b-propeller domain (residues 4–953). It raises a possibility that FBB18 may directly interact with this region, which could be tested in future studies. Interestingly, the abundance of IFT dynein was increased around 3 fold in the mutant (Table 1), suggesting that its folding/stability does not depend on FBB18. This is consistent with a previous report in which DNAAF is not involved in the folding of cytoplasmic dynein [39]. This increase in abundance may reflect a feedback control due to no or short cilia in the mutant. Taken together, FBB18 differentially regulates the stability of dynein HCs of both ODAs and IDAs.

## FBB18 physically associates with almost all axonemal dyneins

Next, we determined which dynein heavy chain species might physically associate with FBB18. To this end, *FBB18*:*YFP-TG* was expressed in *fbb18-1*. The transgenic strains restored the wild-type ciliary phenotypes. Cell lysates of the rescued cells were subjected to immunoprecipitation with anti-GFP antibody, wild-type cells as a control. (S3 Fig). Mass spectrometry analysis of the immunoprecipitates showed that 14 out of a total of 15 axonemal dyneins were associated with FBB18 (Table 2).

To learn whether FBB18 associates with dynein HCs during or after their translation, we examined the identified peptides of DHCα because its stability is severely affected by the loss of FBB18. We found that several identified peptides are located within the C-terminal region of DHCα (S4 Fig), suggesting that FBB18 may mediate folding of completely translated dynein HCs though we could not exclude the possibility that it also functions during dynein translation. We also examined the accessory components of the outer and inner arm dyneins in the immunoprecipitation–mass spectrometry data (S1 Table). For outer arm dyneins, none of the accessory components including IC1 and IC2 were enriched with FBB18-YFP. Interestingly,

**Table 2. FBB18 associates with almost all axonemal dyneins.**

| Group | Gene/Protein | Gene ID | *Control* Spectral count (unique peptides) | *FBB18:YFP-TG* Spectral count (unique peptides) |
|---|---|---|---|---|
| ODA components | DHC13/DHCα | Cre03.g145127 | 22 (19) | 63 (46) |
| | DHC14/DHCβ | Cre09.g403800 | 58 (37) | 111 (66) |
| | DHC15/DHCγ | Cre11.g476050 | 15 (12) | 52 (29) |
| IDA components | DHC1/DHC1α | Cre12.g484250 | 15 (14) | 57 (43) |
| | DHC2/dynein d | Cre09.g392282 | 7 (6) | 98 (57) |
| | DHC3 (minor type dynein) | Cre06.g265950 | - | 2 (2) |
| | DHC4 (minor type dynein) | Cre02.g107350 | - | 11 (1) |
| | DHC5/dynein b | Cre02.g107050 | - | 14 (4) |
| | DHC6/dynein a | Cre05.g244250 | - | 15 (13) |
| | DHC7/dynein g | Cre14.g627576 | - | 4 (1) |
| | DHC8/dynein e | Cre16. g685450 | 1 (1) | 20 (13) |
| | DHC9/dynein c | Cre02.g141606 | - | 252 (106) |
| | DHC10/DHC1β | Cre14.g624950 | 3 (3) | 34 (30) |
| | DHC11 (minor type dynein) | Cre12.g555950 | 4 (3) | 11 (10) |
| | DHC12 (minor type dynein) | Cre06.g297850 | - | - |

Immunoprecipitates with anti-GFP antibody from *fbb18-1* cells expressing *FBB18:YFP-TG* or wild-type cells as control were analyzed by mass spectrometry. Please note, except for the minor dynein DHC12, all the other dyneins were enriched more than 2 fold in the immunoprepitate of *FBB18:YFP-TG*.

only three but none of the other subunits of IDA were enriched. IC140 and IC97 are components of double-headed dyneins while p28 is a component of dynein a, c, and d [1]. Taken together, these data suggest that FBB18 participates in the folding/maturation of most of the DHCs and likely some of the IDA accessory components before they are assembled into large dynein complexes for transporting into cilia though we do not exclude other possibilities.

## FBB18 is in association with several DNAAFs, HSP90 and RuvBL1/2 of the R2TP complex

DNAAFs often form a complex with other DNAAFs, R2TP and chaperones [6, 23, 24]. To learn the protein factors that associate with FBB18 and function in dynein preassembly, we searched known DNAAFs, R2TP components and chaperones in the immunoprecipitates as described above. 16 out of 18 designated DNAAFs are conserved in *Chlamydomonas* [2]. 10 of 16 conserved DNAAFs including FBB18, chaperones HSP90 and RuvBL1/2 of the R2TP complex were identified (Table 3). These data suggest that FBB18 forms a network(s) with these factors to confer its function in axonemal dynein preassembly.

The other six DNAAFs that were not found to be associated with FBB18 include DNAAF7, 9, 12–14 and 18 (Table 3). DNAAF7/ZMYND10 is involved in stabilization of IC1/2 [14] though different results have been reported [40]. DNAAF9/Shulin functions in packaging outer arm dynein complexes for ciliary targeting while DNAAF12/LRRC56 and DNAAF18/ODA16 are involved in maturation and dynein transport, respectively [9, 12, 13]. Thus, these data suggest that FBB18 functions in the chaperoning steps of axonemal dynein HCs but not in IC1/2 stabilization and in maturation/transport of the dynein complexes. Interestingly, the other two DNAAFs that were not associated with FBB18 are DNAAF13/SPAG1 and DNAAF14/MOT48, a PIH domain containing protein. R2TP complexes consist of RuvBL1/2, SPAG1 or RPAP3 and one of the PIH domain containing proteins [16]. SPAG1 or RPAP3 interacts with the PIH containing proteins in the R2TP complex. RPAP3, the paralog of

**Table 3. Factors associated with FBB18 that are involved in dynein preassembly.**

| Proteins | Gene ID | Control Spectral count (unique peptides) | FBB18-YFP Spectral count (unique peptides) |
|---|---|---|---|
| DNAAFs that are associated with FBB18 | | | |
| DNAAF1/DAU1/ODA7 | Cre01.g029150 | - | 11 (6) |
| DNAAF2/ PF13 | Cre09.g411450 | - | 7 (5) |
| DNAAF3/ PF22 | Cre01.g001657 | - | 28 (7) |
| DNAAF4/ DYX1C1/ PF23 | Cre11.g467560 | - | 3 (3) |
| DNAAF5/ HEATR2 | Cre09.g395500 | 44 (21) | 94 (25) |
| DNAAF6/ TWI1/ PIH1D3 | Cre07.g335800 | - | 9 (5) |
| DNAAF10/WDR92 | Cre16.g672600 | - | 3 (3) |
| DNAAF11/MOT47/ LRRC6 | Cre17.g739850 | 1(1) | 5 (4) |
| DNAAF16/FBB18/CFAP298 | Cre16.g688450 | - | 582 (31) |
| DNAAF17/FBB5/ C11orf70/CFAP300 | Cre12.g556300 | 3 (3) | 37 (14) |
| DNAAFs that are not associated with FBB18 | | | |
| DNAAF7/ZMYND10 | Cre08.g358751 | 19(7) | 12(6) |
| DNAAF9/Shulin/C20orf194 | Cre11.g467556 | 1(1) | 1(1) |
| DNAAF12/LRRC56/ODA8 | Cre01.g043650 | - | - |
| DNAAF13/SPAG1 | Cre12.g556100 | 1(1) | (1) |
| DNAAF14/MOT48 | Cre10.g467050 | - | - |
| DNAAF18/DAW1/ODA16/WDR69 | Cre04.g216902 | - | - |
| RPAP3 | Cre02.g084900 | - | - |
| Chaperones and co-chaperones | | | |
| RuvBL1 | Cre09.g414050 | 9 (5) | 220 (25) |
| RuvBL2 | Cre10.g442700 | 7 (5) | 188 (29) |
| HSP90A | Cre09.g386750 | 17 (9) | 48 (17) |
| HSP90C | Cre12.g514850 | - | 8 (6) |
| HSP70E | Cre16.g677000 | 3 (2) | 8 (3) |

Immunoprecipitates from cells expressing *FBB18*:*YFP-TG* or control cells with anti-GFP antibody were analyzed by mass spectrometry. Known dynein preassembly factors, R2TP components and chaperones were searched in the data set and their abundances are presented. Please note that the homologues of DNAAF8 and 15 are not present in the *Chlamydomonas* proteome. RPAP3 forms a R2TP complex with DNAAF14 and RuvBL1/2 and is thus being considered as a possible DNAAF (PMID: 30428028).

SPAG1, was also not identified (Table 3). Previously, we have shown that MOT48 forms a R2TP complex with RuvBL1/2 and RPAP3 [23]. Thus, these data suggest that FBB18 does not associate with a R2TP or its like complex.

## Deciphering the interaction map of the FBB18-associated DNAAFs

Given the association of FBB18 with other DNAAFs revealed by affinity purification, it is intriguing how these factors are organized. Based on previously published data and the data obtained in the current study (see below), we propose a model for their interactions in *Chlamydononas* (Fig 4A). The solid lines indicate interactions derived from previous studies while dashed lines show the interactions revealed by the current study. FBB18 directly interacts with WDR92, ODA7 and LRRC6 as shown by yeast two-hybrid (Y2H) and/or in vitro pull-down assays [23, 32]. LRRC6 directly associates with RuvBL2 in Y2H assay [22]. DYX1C1 directly interacts with either of the two PIH proteins PF13 and TWI1 as shown by LUMIER interaction assay [16], and PF13 but not TWI1 interacts directly with FBB5 as shown by the Y2H and co-immunoprecipitation assays [41].

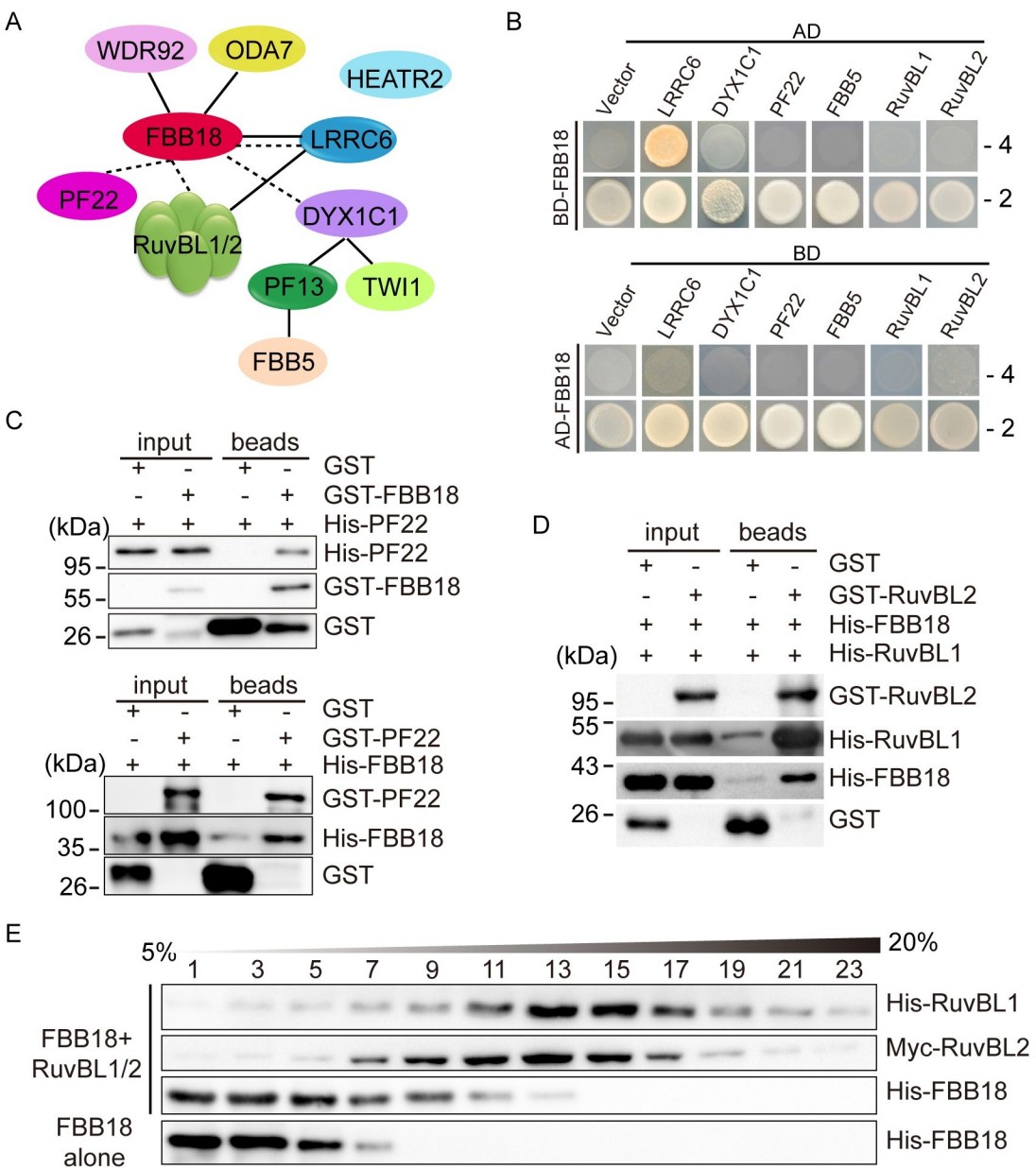

**Fig 4. An interacting network(s) of FBB18 with other DNAAFs.** (A) A model of molecular interaction map of FBB18. Dotted and solid lines represent interactions revealed by current and previous studies, respectively. (B) Y2H assays show direct interaction of FBB18 with LRRC6, DY1X1C1 but not the other proteins tested. Yeast cells that were transformed with each pair of constructs were grown under selection media lacking leucine, tryptophan, histidine, and adenine (-4) or leucine and tryptophan (-2). Empty AD or BD vectors were used as controls. (C) Reciprocal pull-down assays show interaction between FBB18 and DNAAF3/PF22. Bacterial expressed proteins were mixed as indicated and pulled down with glutathione beads followed by immunoblotting with GST and His antibodies, respectively. GST was used as a control. (D) Pull-down assays show interaction between FBB18 and RuvBL1/2. An assay similar to (C) was performed. GST and His antibodies were used for immunoblotting, respectively. (E) Sucrose gradient assay shows that FBB18 forms complexes with RuvBL1/2. Bacterial expressed proteins as indicated were subjected to sucrose gradient analysis followed by immunoblotting with His or Myc antibodies.

Previously published data could not place PF22, HEATR2, DYX1C1 in the above mentioned network associating with FBB18. We first considered possible direct interaction of FBB18 with PF22, DYX1C1, RuvBL1/2 or FBB5 due to their relative high abundance in the

FBB18-YFP immunoprecipitates. HEATR2 was not considered because of high spectral counts in the control immunoprecipitation and only ca. 2 fold enrichment (Table 3). In Y2H assays, FBB18 interacted with LRRC6, confirming a previous study [22] and DYX1C1, but not with PF22, RuvBL1, RuvBL2 and FBB5 (Fig 4B). PF22 and RuvBL1/2 are among the most enriched proteins associating with FBB18 (Table 3). Given the limitations of the Y2H assays, we employed pull-down assays with bacterial expressed proteins. FBB18 and PF22 interacted in reciprocal pull-down assays (Fig 4C). FBB18 interacted with RuvBL1/2 complexes (Fig 4D). To further demonstrate the interaction between FBB18 and RuvBL1/2, bacterial expressed FBB18 and RuvBL1/2 were mixed and analyzed by sucrose gradient and immunoblot analysis (Fig 4E). A portion of FBB18 co-migrated with RuvBL1/2, confirming their interactions. Thus, we have provided an interacting network(s) of FBB18 with other DNAAFs. However, given a small protein like FBB18, how can it directly interact with several DNAAFs? One possibility is that these interactions may not occur simultaneously.

## Discussion

### Regulation of ciliary assembly and length by the assembly of axonemal dynein arms

The null mutant of FBB18 exhibits defects not only in dynein arm assembly but also in ciliogenesis for having short or no cilia. This phenomenon has been reported in *Chlamydomonas* mutants of several other DNAAFs including DNAAF10/WDR92 [23, 39], DNAAF14/MOT48 [20], DNAAF2/PF13 [42, 43], DNAAF4/DYX1C1/PF23 [43, 44] and DNAAF3/PF22 [7, 45]. Furthermore, a double mutant of *pf13/mot48* that causes sever loss of outer and inner dynein arms exhibits much shorter cilia than either single mutant [27]. Thus, it suggests that axonemal dynein arms play a role in ciliary assembly and length control, which is supported by studies of mutants of dynein motors. Complete loss of ODAs in an IC1 mutant (*oda9*) or a DHCγ mutant (*pf28*) does not or slightly reduce ciliary length [45, 46]; the same is true for single loss of several inner arm dyneins [47]. However, double mutants with complete loss of outer dynein arms and individual inner arm species exhibit significantly shorter cilia [45, 47]. Thus, the occupancy of dynein arms in the ciliary axoneme facilitates the stabilization of axoneme and controls ciliary length. This view is supported by the data in which knockout of a tubulin polyglutamylase in a *wdr92* mutant partially restores ciliary length [39]. In several DNAAF mutant patients, the sperm flagellar length is also significantly reduced [48]. However, the length of motile cilia in mammals may not be controlled by the presence of dynein arms. Knockout of DNAAF7/ZMYND10 or DNAAF11/LRRC6 in mouse does not change ciliary length of motile cilia [10, 40]. Thus it appears that the stability of the motile ciliary axoneme may be mainly controlled by other microtubule binding proteins and/or tubulin post-translational modifications [49, 50].

### FBB18 is involved in chaperoning axonemal dynein HCs at the early steps of dynein preassembly

FBB18 was proposed to function in stabilization of dynein arms in cilia and/or in ciliary targeting of dynein complexes [31]. Our data demonstrate that it likely acts at the early steps of axonemal dynein preassembly in chaperoning dynein HCs before formation of dynein holocomplexes. First, FBB18 is predominantly localized in the cytoplasm, similar to what has been described in rat tracheal cells and KV of zebrafish embryos [31, 32]. Second, loss of FBB18 causes instability of multiple axonemal dynein HCs. Third, FBB18 is associated with 14 out of a total of 15 axonemal dynein HCs. None of the accessory components of the outer arm

dyneins including IC1/IC2 were associated with FBB18. Except for the three accessory components (IC140, IC97 and p28), the other accessory subunits of the inner arm dyneins were not associated with FBB18. Fourth, DNAAFs involved in dynein maturation (DNAAF12/LRRC56) [9], packaging (DNAAF9/Shulin) [12] and transport (DNAAF18/ODA16) [13] are not associated with FBB18. Taken together, we propose that FBB18 with its associated DNAAFs is involved in the folding of dynein HCs and likely the three accessory components as well before a holo-complex of dyneins is formed.

## The specificity of DNAAF in chaperoning axonemal dynein HCs

One feature for most of the DNAAFs is that its loss causes severe instabilities of axonemal dynein HCs. To learn the specificity of each DNAAF towards folding/stability of axonemal dyneins, it requires examination of all the dyneins present in particular ciliated cells or organisms. Traditional TEM microscopy, immunoblotting using a few dynein antibodies or even cryo-electron tomography without molecular resolution would not provide such a detailed information. Analysis of dyneins from whole cell sample by mass spectrometry would be an attractive method [40]. By using comparative proteomics with $^{15}$N labeling, we have detected 11 out of a total of 15 axonemal dynein species, the other four undetected species are minor type dyneins [3]. The abundance of all detected dyneins except for DHCγ was reduced with different degrees in *fbb18-1* mutant. Given that FBB18 physically associates with all of these dyneins, it is proposed to participate in the preassembly of all these dyneins. In a comparative proteomics of mouse testes samples, 9 out of 13 mammalian axonemal dynein species were detected with reduced abundance in *zmynd10* null mouse [40]. This study combined with our current study suggest that DNAAFs may have a plasticity nature and have broad functional specificity. Each DNAAF may contribute to the assembly of multiple dyneins with differential preference. Thus, mutation in each DNAAF would cause different outcomes in the instability of dyneins. However, other possibilities exist. The fact that FBB18 associates with almost all of the HCs but has differential regulation of their stability suggests that FBB18 may not directly interact with all of them. It may serve as a part of a network of DNAAFs to facilitate dynein folding. Disruption of the network may result in differential folding of the dyneins, leading to differential instability.

## Formation of multiple DNAAF complexes during folding and stabilization of axonemal dyneins

In the immuoprecipiates of FBB18, a total of 10 DNAAFs have been found including FBB18. Using Y2H, pull-down and other biochemical assays combined with previous literatures, we have proposed an interacting network(s) of these DNAAFs. These DNAAFs may at least form two distinct complexes. FBB18 interacts with DNAAF4/DYX1C1, which interacts with either PF13 or TWI1, both are PIH domain containing proteins [16, 17, 20, 27]. A third PIH domain containing protein in *Chlamydomonas* is DNAAF14/MOT48 [20]. Previously, we found that MOT48 forms a R2TP like complex with RuvBL1/2 and RPAP3 [23]. SPAG1 is a paralog of RPAP3, which can also form a R2TP like complex [16]. Interestingly, MOT48, SPAG1 and RPAP3 all were not found to associate with FBB18. Thus, it suggests that a R2TP like complex exists independently of FBB18. DNAAF7/ZMYND10 involves stabilization of IC1/IC2 and axonemal dynein HCs in mammals [14, 40]. Though mutants of ZMYND10 in *Chamydomonas* have not been reported, its upregulation during ciliogenesis suggests that it may also function in axonemal dynein preassembly in *Chlamydomonas* [36]. We found that ZMYND10 was not associated with FBB18. ZMYND10 forms complexes with a few DNAAFs in mammalian cells [14]. Thus, it indicates that a ZMYND10 interacting module may also exist independently

of FBB18 in *Chlamydomonas*. Taken together, in *Chlamydomonas*, at least 4 DNAAF complexes may be formed simultaneously to mediate folding of axonemal dyneins. It may be argued that most of the DNAAFs forms a loose network, which breaks apart during our immunoprecipitation. Though this is likely, three PIH domain containing proteins in *Chlamydomonas* and four in zebrafish are all involved in axonemal dynein stability [21, 27] and it is expected that each protein forms a discrete complex. These complexes may contribute to the folding of the same dynein species with different capacities, which can explain why mutations of different DNAAFs cause differential regulation of the same dynein species. For example, PIH1D1, KTU, and Twister all contribute to the assembly of inner arm dynein c [21]. Similar conclusions are also drawn in *Chlamydomonas* by systematic studies of PIH containing proteins [27]. Combined with discussion in the last section, we propose the rule governing the folding of dyneins by DNAAFs include two parts: each DNAAF may recruit multiple dyneins and each dynein may interact with more than one DNAAFs. This could explain the defects in one DNAAF lead to instability of multiple dyneins and the stability of one dynein is affected by more than one DNAAFs.

## FBB18 associates with RuvBL1/2 but not a R2TP complex

R2TP complex is a HSP90 co-chaperone involved in folding and assembly of large protein complexes [18]. Recent evidences show that this complex is also involved in preassembly of axonemal dyneins [23, 24]. The conventional R2TP complex in mammal consists of RuvBL1/2, RPAP3 and PIH1D1. The TPR domains in RPAP3 interacts with HSP90 while PIH1D1 recruits chaperone clients [18]. Unconventional R2TP complexes can be formed by substitution of RPAP3 with its paralog SPAG1 or by substitution of PIH1D1 with its paralogs PIH1D2 or DNAAF2/KTU [16]. In *Chlamydomonas*, we previously found that MOT48 forms a R2TP complex with RuvBL1/2 and RPAP3 [23]. Interestingly, our current data showed that FBB18 directly interacted with RuvBL1/2 while RPAP3, SPAG1 or MOT48 was not associated with FBB18, which suggests that RuvBL1/2 could function independently of formation of a R2TP complex. RuvBL1/2 belong to AAA+ superfamily of ATPases. RuvBL1/2 independently of R2TP complex formation have been shown to be involved in the assembly of many protein complexes including chromatin remodeling complexes, and have been proposed to be chaperones [28]. Thus, RuvBL1/2 are likely chaperones involved in axonemal dynein preassembly or serve as a molecular hub to organize DNAAFs.

## Materials and methods

### Strains and culture

*Chlamydomonas reinhardtii* wild-type strain *21gr* (mt+; CC-1690) was obtained from the *Chlamydomonas* Resource Center (University of Minnesota, MN). *Chlamydomonas* mutants *oda1*, *oda5*, and *pf22* were kindly provided by Dr. David Mitchell. Cells were routinely cultured in liquid M or TAP medium as indicated at 23°C with aeration under a 14:10h light/dark cycle.

### Insertional mutagenesis, gene cloning and transformation

Insertional mutants were generated by electroporation of a DNA fragment containing *AphVIII* expressing cassette, which confers the transformants with paromomycin resistance [51, 52]. RESDA PCR combined with DNA sequencing was used to identify the genes with foreign DNA insertion [53]. *FBB18* gene including an ~1.5-kb fragment upstream of the start codon was cloned by PCR. The gene was tagged with 3xHA or YFP followed by a rubisco terminator.

The resulting constructs were cloned into a modified vector pHyg3 harbouring a hygromycin B resistance gene [54]. The constructs for the truncated mutants as indicated in the text were generated by PCR and cloning. The final constructs were linearized with KpnI for transformation.

## Cell fractionation, ciliary regeneration, ciliary length measurement, and DIC cell imaging

Previous published methods were followed [55]. Briefly, cells were deciliated by pH shock followed by isolation of cell bodies and cilia or allowed for cilia regeneration. For cilia fractionation, isolated cilia were dissolved into HMDEK buffer containing 0.5% NP40 followed by centrifugation to separate the membrane/matrix and the axenemal fractions. For measurement of ciliary length or scoring ciliary phenotype, cells were fixed with 1% glutaraldehyde and imaged by DIC microscopy using Zeiss Axio Observer Z1 equipped with a CCD camera (QuantEM512SC, Photometrics, USA). Cell images were processed with Photoshop and Illustrator (Adobe, USA) and ciliary length was measured using ImageJ (NIH, USA). 50 cilia were measured to determine the mean length of cilia.

## Ciliary beat analysis

Cells were imaged with a Nikon microscope (A1RSi) equipped with an ORCA-flash 4.0 camera (Hamamatsu, Japan). Phase-contrast optics and a 40x/0.65 NA objective were used. Images were captured at 400 fps and analyzed with HCImage Live software (Hamamatus, Japan). Beat frequency data were analyzed using Image J and graphed by GraphPad Prism 5 (GraphPad, USA). The average frequency was determined by analyzing 39 cells.

## Primary antibody

Details for the primary antibodies for immunoblotting (IB) and immunofluorescence (IF) can be found in S2 Table. Please note, a rabbit polyclonal antibody of FBB18 was raised against a full-length polypeptide and affinity purified. DHCα and DHCβ antibodies were kindly provided by Dr. David Mitchell (SUNY Upstate Medical University), dynein b, c and e antibodies by Dr. Toshiki Yagi (Prefectural University of Hiroshima, Japan) and FMG1 antibody by Dr. Robert Bloodgood (University of Virginia).

## SDS-PAGE, immunoblotting, immunoprecipitation and mass spectrometry

The methods for SDS-PAGE, immunoblotting and immunoprecipitation were essentially as described previously [56, 57]. For immunoprecipitation, $1 \times 10^9$ of *fbb18-1* rescue cells expressing FBB18-YFP were used with wild-type cells as a control. FBB18-YFP was pulled down with anti-GFP rabbit mAb conjugated agarose beads (Engibody, USA). The immunoprecipitates were subjected to SDS-PAGE and the gel slices were analyzed by mass spectrometry in the Core Facility of the Center for Biomedical Analysis (Tsinghua University). The mass spectrometry data presented for the immunoprecipitation are from one experiment.

For comparative proteomic analysis of axonemal dyneins, *fbb18-1* mutant and rescued cells were labelled with $^{15}$N by culturing the cells in medium containing $^{15}NH_4$$^{15}NO_3$, respectively. Equal amounts of proteins from $^{15}$N labelled *fbb18-1* cells and unlabeled rescued cells were mixed, or vice versa. 200 μg proteins were subjected to SDS-PAGE (7.5%) followed by coomassie blue staining. The gel part above a 310 kD protein marker was sliced for mass spectrometry analysis. pFind algorithm was used for data analysis with default setting [58].

### IF and thin-section electron microscopy

A previously published protocol for IF was followed [57]. The secondary antibodies used were Texas red-conjugated goat anti-mouse IgG (1:400) and Alexa Fluor 488-conjugated goat ani-rat IgG (1:400; Molecular Probes). The samples were viewed on a Zeiss LSM780 META Observer Z1 Confocal Laser Microscope. Images were acquired and processed by ZEN 2009 Light Edition (Zeiss) and Photoshop (Adobe). Analysis for electron microscopy followed published methods [52, 59]. The samples were imaged on an electron microscope (H-7650B; Hitachi Limited) equipped with a digital camera (ATM Company).

### Pull-down assays

Bacterial expressed proteins were used for pull-down assays. cDNAs for *Chlamydomonas* genes were cloned into pGEX-6P-1, pET28a or pETDuet-1 vectors and expressed in *E. coli* Rosetta (DE3). Of note, His-RuvBL1 and GST-RuvBL2 or GST were co-expressed. The cell lysates were mixed before pull-down using GST glutathione beads (Smart LifeSci, China).

### Sucrose gradient assay

A 2 ml of sucrose linear gradient (5–20%) was made [60]. 100–200 μl protein samples were loaded on top of the gradient followed by centrifugation at 200,000g for 4 h at 4°C (Beckman MAX-XP, rotor TLS-55). The fractions were subjected to immunoblotting. For co-migration assay of RuvBL1/2 and FBB18 from bacterial expressed proteins, co-expressed RuvBL1/2 were mixed with FBB18 before loading onto the sucrose gradient.

### Yeast two-hybrid assays

*Chlamydomonas* cDNA of each gene was cloned into the prey vector pGADT7 (AD) and the bait vector pGBKT7 (BD), respectively. Each pair of constructs as indicated were co-transformed into the yeast strain AH109 following a previous method [61]. Transformants were grown at 30°C for 2–3 days on selection media (SD, -Leu, -Trp, -His, -Ade or SD, -Leu, -Trp) (Clontech, USA). Empty vectors were used as controls.

### Statistical analysis

Data were presented as mean ± SD. Statistical significance was performed by using unpaired two-tailed Student's t-test analysis. $P < 0.01$ was considered to be statistically significant. ***$P < 0.0001$.

## Supporting information

**S1 Table. The mass-spectrometry data of the accessory subunits of dynein from FBB18-YFP immunoprecipitation.** Immunoprecipitates with anti-GFP antibody from *fbb18-1* cells expressing *FBB18-YFP* or wild-type cells as control were analyzed by mass spectrometry. Please note, only the subunits highlighted in red were identified both with more than 2 unique peptides and a more than two-fold increase in the spectral count (sample versus control). (DOCX)

**S2 Table. Primary antibodies used in this study.** (DOCX)

**S1 Fig. Analysis of functional domains of FBB18.** (A) Domain organization of FBB18 and its truncated proteins. The coiled-coil domain and DUF2870 are deleted in FBB18$^{\Delta25-53}$ and FBB18$^{\Delta177-277}$, respectively. FBB18$^{\Delta252-277}$ mimics a human mutation that results in a C-

terminus truncated protein. Numbers denote the residue positions. (B) Immunoblot analysis of the expression of the deletion mutants. Deletion constructs with a wild-type construct as a control were expressed in *fbb18-1* followed by immunoblotting. Wild-type (WT) and *fbb18-1* cells were also used as controls. Arrow and arrowhead mark the truncated proteins FBB18$^{\Delta25-53}$ and FBB18$^{\Delta177-277}$, respectively. Band marked by asterisk indicates non-specific band.
(TIF)

**S2 Fig. Analysis of the abundance of FBB18 in whole cell and cilia samples of various motility mutants.** Whole cells (A) or isolated cilia (B) from cells as indicated were subjected to immunoblotting with the indicated antibodies. Please note, the ciliary abundance of FBB18 varies with the mutants. We failed to observe a correlation between the ciliary abundance of FBB18 with the phenotype of impaired cilia motility.
(TIF)

**S3 Fig. Immunoprecipitation of FBB18-YFP.** Cells expressing FBB18-YFP with wild type cells as a control were immunoprecipitated with anti-GFP antibody followed by immunoblotting with anti-GFP antibody (A) and SDS-PAGE (B). The gel slices were used for mass spectrometry analysis.
(TIF)

**S4 Fig. The amino acid sequence of DHCα and its identified unique peptides in FBB18-YFP immunoprecipitation.** Residues highlighted with red or blue are the peptides identified by mass spectrometry.
(PDF)

**S1 Video. Time-lapse movies showing the motility of wild-type cells.** Cell images were captured at 400 *fps* and the movie is played at 10 *fps*.
(AVI)

**S2 Video. Time-lapse movies showing the motility of fbb18-1 cells expressing FBB18Δ252–277.** Cell images were captured at 400 *fps* and the movie is played at 10 *fps*.
(AVI)

**S3 Video. Time-lapse movies showing the motility of fbb18-1 cells expressing full-length FBB18 tagged with 3xHA.** Cell images were captured at 400 *fps* and the movie is played at 10 *fps*.
(AVI)

## Acknowledgments

We thank Drs. David Mitchell, Toshiki Yagi and Robert Bloodgood for providing antibodies and/or strains and the Core Facility of the Center for Biomedical Analysis (Tsinghua University) for assistance on cell imaging and mass spectrometry analysis.

## Author Contributions

**Conceptualization:** Junmin Pan.

**Data curation:** Limei Wang, Xuecheng Li, Guang Liu.

**Formal analysis:** Limei Wang, Xuecheng Li, Guang Liu, Junmin Pan.

**Investigation:** Limei Wang, Xuecheng Li.

**Methodology:** Limei Wang, Xuecheng Li, Guang Liu.

**Project administration:** Junmin Pan.

**Resources:** Junmin Pan.

**Software:** Junmin Pan.

**Supervision:** Junmin Pan.

**Validation:** Limei Wang, Xuecheng Li, Guang Liu.

**Visualization:** Limei Wang, Xuecheng Li, Guang Liu.

**Writing – original draft:** Limei Wang, Junmin Pan.

**Writing – review & editing:** Xuecheng Li, Junmin Pan.

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
