## [Decision Letter · Decision Letter 0]

20 May 2022

Dear Dr Pan,

Thank you very much for submitting your Research Article entitled 'CFAP298 participates in preassembly of almost all axonemal dyneins independent of R2TP complex' to PLOS Genetics.

The manuscript was fully evaluated at the editorial level and by independent peer reviewers. The reviewers appreciated the attention to an important problem, but raised some substantial concerns about the current manuscript. Based on the reviews, we will not be able to accept this version of the manuscript, but we would be willing to review a much-revised version. We cannot, of course, promise publication at that time.  It is important to address the comments about the presence of FBB18/FAP298 in the ciia.  This makes it very different from other DNAAFs and needs to be addressed. 

The Chlamydomonas community has adopted the new nomenclature and I ask that you follow the new rules.

If you decide to revise the manuscript for further consideration at PLOS Genetics, please aim to resubmit within the next 60 days, unless it will take extra time to address the concerns of the reviewers, in which case we would appreciate an expected resubmission date by email to plosgenetics@plos.org.

[LINK]

We are sorry that we cannot be more positive about your manuscript at this stage. Please do not hesitate to contact us if you have any concerns or questions.

Yours sincerely,

Susan K. Dutcher

Associate Editor

PLOS Genetics

Gregory P. Copenhaver

Editor-in-Chief

PLOS Genetics

Nomenclature

• You need to follow the nomenclature rules at the Chlamydomonas Resource Center. For example, a transgene should be FAP298:HA-TG. You need to indicate it is a transgene and not an edited gene. https://www.chlamycollection.org/resources/tools/gene-nomenclature/

• Genes should NEVER include Cr in the name.

• Do not use superscripts, but give them allele names. (see link to nomenclature)

• This gene has been named in Phytozome in 2005 and , it has been published as FBB19 (My choice for the Chlamy gene). It is also known as kurly, C21orf59 and finally as CFAP298. No matter what, it is not CFAP298.

• CFAP298 is the human protein name and not the Chlamydomonas name. You must use it, it is FBB18 or FAP298.

Introduction and Discussion points

• The manuscript states that DNAAF7 stabilizes IC1/IC2. The authors should look at the work from Mali (REF 47).

• Patients have laterality defects rather situs inversus as they are patients with heterotaxy. Laterality is a more general term.

• Wild-type should have a hyphen when used as an adjective (line 125 as one example).

• The American Society of Human Genetics recommends that changes in patients be referred as variants and not as mutations. (one example on line 171).

• Pleases avoid flagella and deflagellation. See line 190 on page 8.

Experimental Questions

• The FAP298:YFP-TG construct is never characterized but is used in several experiments. It must be characterized. The authors should at least look at cilia numbers and lengths. They should characterize at least three independent transformants.

• The ODA and IDA protein levels should be quantitated.

• The manuscript states that DNAAF7 stabilizes IC1/IC2. The authors should look at the work from Mali (REF 47).

• The authors state that cilia are isolated with 0.5% Non-Idet. I would guess that they have axonemes and not cilia.

• In the previous characterization of FBB18 in the cilia, it was stated that the level is low. Please compare your results to the published data.

• The proteomics and the blots do not agree very well. Please discuss

Reviewer's Responses to Questions

**Comments to the Authors:**

Reviewer #1: Defects in CFAP298 (aka C21orf59 and FBB18) have previously been shown to affect dynein assembly/ciliary motility in zebrafish, humans and planaria; human patients exhibit primary ciliary dyskinesia phenotypes. Biochemical studies using Chlamydomonas found the protein both in the cytoplasm and the ciliary matrix. In this study, the authors identify and characterize a Chlamydomonas null mutant in CFAP298. They find it is present mainly in the cytoplasm but that it is also readily detectable in cilia. The mutant exhibits no or short cilia and proteomics/pull downs identify associations with dynein heavy chains (HCs) and several other factors known to be involved in the cytoplasmic preassembly of axonemal dyneins. They find that CFAP298 associates with the RuvBL1/2 ATPases but they find no evidence for association with other components (scaffolds and PIH proteins) of R2TP complexes and its variants.

Comment to the Journal

The figures provided as part of the manuscript pdf are very low quality. I can only assume that this reflects the automated manuscript processing system that generated the pdf as the supplemental data file figures provided directly by the authors are of high quality. Providing such low quality figures to reviewers does authors a great disservice.

Comments to the Authors

1. The abstract (and many other places) should indicate that the ODA/IDA loss is partial in the cfap298 mutant and that the amounts of some major HCs are barely affected. Also, the abstract refers to “new mechanisms of axonemal dynein preassembly” but no mechanistic detail is actually provided.

2. The authors show that the alpha HC is missing and the amount of beta HC in cfap298 cilia is reduced somewhat. Based on this and similar data for several inner arms they state that CFAP298 is “required for the assembly of both ODA and IDA in cilia” (lines 157-158) – considering many ODA/IDAs still assemble this seems a rather strong statement as the effect is clearly not “all or none” but is rather much more nuanced.

3. The observation that assembly of the alpha HC particularly depends on the CFAP298 DUF2870 domain is interesting as the alpha HC (which is closely related to the unaffected beta HC) is unusual in having a completely different N-terminal region compared to all other Chlamydomonas HCs. Although not a requirement, it might be interesting to test whether CFAP298 interacts specifically with this alpha HC region that consists of kelch domains and several Ig folds.

4. Although CFAP298 is said to associate with the beta and gamma ODA HCs, its loss appears to have little/no effect on their cytoplasmic assembly as they are present in the mutant at ratios of 0.9 and 1 compared to wildtype (it is quite possible that the small reduction in beta HC amount might reflect the total loss of its binding partner the alpha HC rather than a direct effect of CFAP298 loss). As these (beta and gamma) are 2 of the 3 most abundant axonemal HCs in Chlamydomonas, if CFAP298 is central to HC folding/stability why are they unaffected? From my understanding, the IP (table 2) was from cell extracts and so if CFAP298 binds alpha HC (which is greatly affected by cfap298 loss), they would also IP beta and gamma which are tightly bound to alpha even though CFAP298 may not be actually binding them directly at all. Similarly, if CFAP298 associates with one HC of the I1/f dynein one might expect to pull down the second HC as well even though it may not interact directly. Thus, this needs clarifying. Care should be taken not to imply that CFAP298 is a DNAAF for all HCs, which it clearly is not. Many sentences mentioning this concept are sweepingly broad (e.g. lines 333-334 and elsewhere) and it should be made clear they do not apply to all axonemal dynein HCs equally. Perhaps it might help if the authors could identify or discuss some common feature(s) of the HCs most dramatically affected compared to those that are not. What special feature of these HCs might CFAP298 affect?

5. In the IPs (table 2), where in the HCs do the identified unique peptides derive from? Do they include C-terminal peptides in which case one might predict that CFAP298 is associating with fully formed dyneins (i.e. post chaperone requiring) – how would this affect their interpretation of CFAP298 function? Also, the authors should detail what other dynein components (if any) were found in these IPs as this again directly impacts on the state of the dyneins with which CFAP298 is associating.

6. The authors observe that loss of cfap298 leads to an increase in the IFT dynein and suggest that its folding/stability does not depend on CFAP298. The observation that no DNAAFs (including CFAP298) have been reported to adversely affect cytoplasmic or IFT dynein assembly was made previously (see ref #39) and should be acknowledged.

7. The authors state that CFAP298 participates in dynein folding in cytoplasm based on interactions with several DNAAFs and contrast this with previous proposals that it might target dyneins to cilia or stabilize them within cilia as CFAP298 was found in cilia. As the authors also demonstrate and confirm the presence of CFAP298 in isolated cilia and the cilia membrane/matrix fraction what role do they predict it plays there? Presumably not an early step in preassembly? As many DNAAFs are totally absent from cilia this is very different and although the cilia signal in Fig 3C is quite low, a very robust signal is shown in Fig 3B. Perhaps including Coomassie stained gels would allow the gel loadings to be better appreciated. I think at several places in the manuscript there is an assumption that a role in cytoplasmic folding/stability precludes involvement in other activities or locations, but without providing any data to rule them out. This can be readily dealt with by rewriting several sections to be less assertive. Considering how long it takes a ribosome to actually synthesize a dynein HC, it is also unclear how “chaperoning” per se can be readily distinguished from many other steps in the pathway to functional dyneins as they are likely cotemporaneous and may occur prior to the completion of HC synthesis.

8. A previous study found more CFAP298 in flagella and cell bodies of several motility mutants compared to wildtype. The data provided here in Fig S2 using a different antibody show less increase and the authors say any differences are not significant (lines 200-202) but on what basis is unclear. Indeed, it does seem under some conditions that increases are actually observed. For example, in Fig S2A expt 1 there would appear to be clearly more CFAP298 in ida1 cells compared to wildtype and similarly in Fig. S2B expt 1 there is more in oda1 and oda5 cilia compared to wildtype (this may also be the case in expt 2 but the blot exposure is very dark). Also in Fig S2B, the amounts in pf20 cilia seem to be consistently much less than other strains. At the least, there does appear to be considerable variability in the signals obtained in the various mutants, which should be directly acknowledged even though the origin of the variation might be uncertain.

9. Fig 3C – why is there no tubulin signal in the whole cell/cell body samples while signal is readily visible in the cell extract blots shown in Fig 1E? Again, Coomassie stained gel/blots would aid interpretation here.

10. Also in Fig 1E, there are several non-specific bands detected by the CFAP298 antibody that are present in all cell extracts. However, in addition to CFAP298 itself, the CFAP298 antibody also detects a band at ~50 kDa in wildtype that disappears in the mutant and reappears in the rescued strain. Is this some other CFAP gene product or splice variant? Is it detected by the HA tag antibody (the WB strip shown is not large enough to tell)? Would it also be included in the IPs shown elsewhere in the paper, and how might this affect the interpretation of protein-protein interactions especially for proteins found in low abundance?

11. Immunostaining in Fig 3D indicates a punctate pattern throughout the entire cell including where I would anticipate the chloroplast, which fills much of the cell, is located. Is this a maximal projection image or ?????, the methods are very unclear on this. Please clarify.

12. The diagram in Fig 4A indicates that CFAP298 interacts directly with 6 different proteins. Given its rather small size, it seems unlikely this could all occur at the same time. Does binding of one or more of these components preclude binding of others? This information might significantly aid interpretation of CFAP298 function.

Reviewer #2: Summary. The authors use multiple complementary methods to analyze the function of CFAP298 in Chlamydomonas. The data from the insertional CFAP298-1 mutant suggests a role with the assembly of specific heavy chain outer dynein arm DHC alpha and inner dynein arm dynein b. For such function, CFAP298 interacts with many other dyneins. I have a few questions and requests for additional information that may improve the manuscript.

Questions

Line 135 and Figure 2C and Figure 3E. If only 23% of the transformants from the rescue had a normal phenotype, how does the level of DHC alpha and dynein b appear identical to the wild type levels?

Line 199 and Figure 3B, S1A. Please discuss the role of CFAP298 in the flagella/cilia. Such location is not consistent with other DNAAFs. Is it possible that there is an additional transport role.

Line 206 and Figure 3D. The figure 3D should be improved. The CFAP298-HA images be improved and especially brightened? It is difficult to see the puncta in green contrasted to black and observe if there is any signal in the flagella. Can the authors co-localize CFAP298 with axonemal dynein motor client cargo? Such experiments would strengthen the difference from prior published studies (ref 31) and support the IP experiments.

Lines 234-237 and Table 1. The explanation for finding more DHC alpha in the CFAP298-1 cells compared to control cells is not obvious. If CFAP298 if functioning to stabilize dynein (Line 214) would it not be likely that the target protein would be degraded? The western blot in of the mutants shows it is not detect in Fig 3E ?

Minor points

Table 1. Please explain and add to methods how were counts normalized for each mass spec sample and how the average ratio determined.

Line 131. Please include the site of the immunogen for the ab to CFAP298 in the main text. It is important for interpreting the data and I could not find that information in the methods or supplements.

Fig 4A. It would be useful to have a key for solid and dashed lines. Some interactions are previously reported interactions and identified in the current manuscript, so both types of lines would be used?

Reviewer #3: This study uses a CFAP298 null mutation in Chlamydomonas to add to biological understanding of axonemal dynein arm assembly, a complex chaperone-driven process. CFAP298 is not well understood after human disease and zebrafish studies and this report adds a useful critique of past papers and the current understanding. Defective CFAP298 is here linked to flagellar motility and length defects plus defects of both outer and inner arm dynein assembly. In constructs over-expressed on a CFAP298-null background, deletion of either the CFAP298 CC or DUF domain leads to protein instability. Over-expression of del252-277 was more useful, mimicking a reported human mutation causing PCD with dynein arm defects. This construct was less able to rescue flagella motility or ODA structures that FL. Mass spec interaction studies of CFAP298 to identify its protein network shed some fresh light on its protein partners and notably showed that CFAP298 is likely to interact with RuvBL1/2 without SPAG1 and RPAP3.

Generally speaking, this study adds information to the field, there are some additional controls that could have been done on protein experiments and some of the English grammar could benefit from review sub editing.

1. Have the authors looked at an alphafold model of CFAP298 and whether the model can tell something about functions of domains of the protein and mutations, in particular how it could fit into a R2TP type of complex.

2. Fig1B, it is unclear how exactly the gene insert affects the CFAP298 protein to create a null allele; e.g., a fusion protein? Early termination?

3. In the case of the human mutation p.Tyr245*, nonsense mutations are generally expected to lead not to an expressed truncated protein, but to nonsense mediated decay and degraded hence no protein and in this sense the Chlamy over expressed delta252-277 may not be a good model of the mutational effect. Is there any evidence for the human mutant protein and whether it is expressed or lost from patient cells?

4. Up to Fig3 the authors talked about delta252-277 but they don’t mention anything about this construct in Fig4. It would be interesting to know how this truncation affects protein-protein interactions shown in Fig4.

5. Fig 2C and 3E cilia versus cells, in the discussion what is the relevant comparison for these similar experiments. If possible, it would be useful to create a summary cartoon of the predicted effect of the loss of CFAP298 on the axonemal dyneins structures.

6. Table 2 and 3 IP data – these experiments pick up a lot of interactions and some are much weaker than others without any subsequent following up of these results for most. Was an empty-YFP construct included to these experiments as a control, can this data be shown in comparison? Are synchronised cultures of Chlamydomonas being compared. The text could do more to acknowledge some of the study limitations.

Can the text clarify whether Table 2 and 3 use the same mass spec data, and how many times was this experiment done, to understand variance in the results. With no follow up coIP studies shown for most of these interactions except the PF22 and RUVBL1/2, this would strengthen the evidence.

7. Discussion – refer to the possible reasons why the difference is seen in expression levels of CFAP298 in Chlamydomonas mutants in FigS1, compared to the past report [31].

8. Punctate staining of CFAP298 in the cytosol – have authors seen evidence that it is present in DyNAPs, if TEM was used. Ref to work of Wallingford group PMID: 33263282.

9. Discussion finishes abruptly – is it suggested from this data that CFAP298 could be in a different type of R2TP like complex, or in a differently structured complex? The notion of multiple R2TO type complexes as set out in reference 16, is the author’s speculation that this is still most likely for CFAP298.

Technical comments, typos:

- Fig1C graph, refers to cell numbers with sizes – how is this important what does it say about phenotype? This graph also needs to show the equivalent data for WT to understand what the normal spread of cell size is.

- What does it mean that cells were ‘hatched from sporangia for unknown reason’ with reference to Fig1, this is not further discussed, what is the relevance?

- Text on Fig1 suggests we should be seeing evidence of a 23% rescued phenotype – what data does this refer to?

- Page 6 of text only refers to Figure 1 for motility/length evidence, this needs to refer to Fig 2 data graphs.

- Fig2B why does the WT lack cross sections containing the expected 9 ODAs.

- Fig2D, these graphs really should show the equivalent cfap298-1 mutant data as well as the rescue data, for cilia length and beat frequency. Technical details seem incomplete – how many times was the experiment done, number of cells tested?

- The Fig2D data could benefit from showing movies of the rescued or non-rescued beating and length.

- Page 6 first referring to CFAP298-HA, this needs to explain this is a FL HA tagged version.

- Fig3A the use of a nonspecific band as a control is clearly unusual and seems unacceptable if that protein is unidentified. What is pdf, a control? Where can we see the wildtype levels of protein, to judge the level of successful rescue?

- Fig3E would benefit from a normalised graph of protein levels compared to CBB.

- Page 10 starts ‘After ascertaining that CFAP298 is a DNAAF etc’, but this is already established with human gene nomenclature DNAAF16 well established and so on. Need to indicate this is a confirmation therefore, or is it the first finding of such.

- Fig4B the criticised Y2H data missing key CFAP298 interactions, this seems better to be moved to supplementary data then.

- Table 1 data could benefit from a graph to show clearly the changed protein abundances.

- Table 3 what is the relevance in R2TP complexes of the two HSP90s A, C, and the HSP70E, which have been associated to DNAAFs before.

- In text and figures ensure DYX1C1 has the final 1 on the end, this is missing a couple of times.

- Bottom of page 12 add reference to Figure 4E here in the text.

- Page 14 typo DNNAFs, check for a few more minor typos. Such as line 351 page 14 attractive, line 384 page 15 axonemal, line 385 loose network.

**Have all data underlying the figures and results presented in the manuscript been provided?**

Reviewer #1: Yes

Reviewer #2: Yes

Reviewer #3: **No: **Mass spec raw data has not been provided

PLOS authors have the option to publish the peer review history of their article (what does this mean?). If published, this will include your full peer review and any attached files.

Reviewer #1: No

Reviewer #2: No

Reviewer #3: No

---

## [Decision Letter · Decision Letter 1]

12 Jul 2022

Dear Dr Pan,

Thank you very much for submitting your Research Article entitled 'FBB18 participates in preassembly of almost all axonemal dyneins independent of R2TP complex' to PLOS Genetics.

The manuscript was fully evaluated at the editorial level and by independent peer reviewers. The reviewers appreciated the attention to an important topic but identified some concerns that we ask you address the minor revisions in a revised manuscript.  There are only a few places where the current Chlamydomonas nomenclature is not used.  The gene is FBB18:YFP-TG or FBB:HA-TG.  Please put the TG in the name of the gene on lines 257 and 779.  Figure 3 does not have a control for the level of contamination for the cytoplasm in the cila prep.  Is the FBB18 signal just cytoplasmic contramination;? Please use the NAB antibody or PFOR antibody to ask about cytoplasmic contamination.  This is key since ciliary localization in not observed in C. elegans or human cells.

We therefore ask you to modify the manuscript according to the review recommendations. Your revisions should address the specific points made by each reviewer.

[LINK]

Yours sincerely,

Susan K. Dutcher

Associate Editor

PLOS Genetics

Gregory P. Copenhaver

Editor-in-Chief

PLOS Genetics

Reviewer's Responses to Questions

**Comments to the Authors:**

Reviewer #1: I think the authors have addressed most of my comments. However, concerning the variable amounts of FBB18 in cilia samples from various mutants, although they state they now acknowledge there is variability, the legend to the supplemental figure S2 still contains the statement “no significant changes in the abundance of FBB18………”. If one looks for example at the pf20 cilia, there is very obviously much less FBB18 than in say oda5 cilia; a similar statement can be made about wildtype vs. oda5. The lower cilia blot is very over exposed but even there the differences in samples are clear. Surely, this is significant and it is also consistent. If not, what difference would the authors consider significant? So I still think the authors are seeing consistent changed levels of FBB18 between at least some of the strains they examined.

I am also not sure I agree with the author’s point concerning the immunostaining which shows a punctate pattern throughout the cell even where the chloroplast should be. The argument is that on fixation the cytoplasm can flow into the damaged chloroplast and thus the antigen appears to be everywhere. If correct, doesn’t this negate the entire point of the experiment, which is to localize FBB18? Shouldn’t every other cytoplasmic antigen show the same pattern in this case? Perhaps a different fixation scheme using an aldehyde rather than methanol should be used to avoid this “flow”.

Reviewer #2: Summary. The authors have submitted a revised text with additional explanations of their data. I have a few suggestions to improve comprehension of the data.

1. The writing and grammar should be improved, particularly in the new (revised) text. The abstract should especially be edited for clarity. For example, Line 22: The sentence confuses the wild type and mutant alleles. I did not see evidence that the mutation of FBB18 localizes to the cytoplasm. I suggest moving cytoplasmic localization to line 20: Here, we show that FBB18, a Chlamydomonas homologue of CFAP298, localizes to the cytoplasm and functions in the folding/stabilization…

2. Results. Lines 240-252 are difficult to follow in the revised form since the reader has to move between the immunoblots in Figure 3 and Table 1 mass spec data. I suggest inserting another reference to (Table 1) at the end of the sentence referring to IFT dynein on line 249.

3. Line 131. Please include the immunogen used to generate the ab to FBB18 in the main text. It is important for interpreting the data and I could not find that information in the methods, figure legend or supplement. Another reviewer asked for that same information. It is not useful to only reference that information.

**Have all data underlying the figures and results presented in the manuscript been provided?**

Reviewer #1: Yes

Reviewer #2: Yes

PLOS authors have the option to publish the peer review history of their article (what does this mean?). If published, this will include your full peer review and any attached files.

Reviewer #1: No

Reviewer #2: No

---

## [Editor Report · Decision Letter 2]

4 Aug 2022

Dear Dr Pan,

We are pleased to inform you that your manuscript entitled "FBB18 participates in preassembly of almost all axonemal dyneins independent of R2TP complex" has been editorially accepted for publication in PLOS Genetics. Congratulations!

Yours sincerely,

Susan K. Dutcher

Academic Editor

PLOS Genetics

Gregory P. Copenhaver

Editor-in-Chief

PLOS Genetics

Comments from the reviewers (if applicable):

**Data Deposition**

http://datadryad.org/submit?journalID=pgenetics&manu=PGENETICS-D-22-00477R2

**Press Queries**

---

## [Editor Report · Acceptance letter]

21 Aug 2022

PGENETICS-D-22-00477R2 

FBB18 participates in preassembly of almost all axonemal dyneins independent of R2TP complex 

Dear Dr Pan, 

We are pleased to inform you that your manuscript entitled "FBB18 participates in preassembly of almost all axonemal dyneins independent of R2TP complex" has been formally accepted for publication in PLOS Genetics! Your manuscript is now with our production department and you will be notified of the publication date in due course.

With kind regards,

Zsofia Freund

PLOS Genetics

On behalf of:
